# Prevalence, Disability and Associated Factors of Playing-Related Musculoskeletal Pain among Musicians: A Population-Based Cross-Sectional Descriptive Study

**DOI:** 10.3390/ijerph17113991

**Published:** 2020-06-04

**Authors:** Rosa Gómez-Rodríguez, Belén Díaz-Pulido, Carlos Gutiérrez-Ortega, Beatriz Sánchez-Sánchez, María Torres-Lacomba

**Affiliations:** 1Physiotherapist in private clinic, 28850 Madrid, Spain; rosagomrod@gmail.com; 2Physiotherapy Department, University of Alcalá, Alcalá de Henares, 28805 Madrid, Spain; 3Medical Statistics Unit, Department of Epidemiology, Hospital Central de la Defensa, 28047 Madrid, Spain; cgutort@oc.mde.es; 4Physiotherapy Department, Physiotherapy in Women’s Health Research Group, Faculty of Medicine and Health Sciences, University of Alcalá, Alcalá de Henares, 28805 Madrid, Spain; beatriz.sanchez@uah.es (B.S.-S.); maria.torres@uah.es (M.T.-L.)

**Keywords:** musicians, occupational disorders, musculoskeletal pain, disability, prevalence, risk factors

## Abstract

*Background:* Playing-related musculoskeletal disorders are the most frequent complaints among instrumental musicians. The aims of this study were: to assess the prevalence of musculoskeletal pain; to evaluate neck, shoulder, and lower back disability; and to determine the associated factors with the presence of musculoskeletal pain among musicians. *Methods:* A population-based, cross-sectional descriptive study was conducted. We selected Spaniard musicians over 16 years old who played a musical instrument for at least five hours per week. They answered the Spanish versions of the Standardised Nordic Questionnaire, the Oswestry Disability Index, Neck Disability Index and Shoulder Pain and Disability Index. Results: We found 94.8% of musicians presented at least one symptomatic region in the last 12 months, and 72.3% in the last seven days. Female musicians (OR 4.38, CI 2.11−9.12), musicians with overweight or obesity (OR 5.32, CI 2.18−12.97), and musicians who play more than 14 h per week (OR 3.86, CI 1.80−8.29)were shown to be a higher risk of suffering musculoskeletal pain. *Conclusions:* Musculoskeletal disorders symptoms are highly prevalent in musicians. The main risk factors related to musculoskeletal disorders symptoms were gender (being female), overweight, obesity, and spending playing more than 14 h a week practicing. This study highlights the need to provide strategies to prevent occupational disabilities among musicians. Further studies are needed to analyse the prevalence of pain in the musician using other sampling methods.

## 1. Introduction

Playing-related musculoskeletal disorders (PRMD) are the most frequent complaints among instrumental musicians. Three diagnoses predominate in the literature: overuse syndrome, entrapment neuropathy and focal dystonia [1]. Pain is the primary playing-related musculoskeletal symptom, but it may be variably described as aching, burning, electrical or pulsating [2].

In 2016, Kok et al. conducted a review of the prevalence of PRMD. The reported point prevalence rates varied from 57% to 68% for musculoskeletal complaints, and from 9% to 68% for playing-related complaints. Playing-related complaints prevalence ranged between 41% and 93% in the last 12 months. No instrumental group showed a greater significant prevalence, although brass had the lowest prevalence of musculoskeletal problems. The number of affected areas varied greatly between studies. In general, the more frequently affected were the neck and shoulders, and the fewer, the elbows. They concluded the importance of using a uniform definition, the same prevalence times and using validated questionnaires to improve the scientific quality of studies on musculoskeletal pain and its associated factors [3].

Regarding disability, Paarup et al., in 2011, determined that 73% of orchestral musicians had an alteration or modification of the way of playing due to musculoskeletal symptoms [4]. Following this review, different studies investigated various risk factors for musculoskeletal problems among musicians, but they present some limitations, such as little information regarding the number of musicians included in the sample [5], or the use of non-validated questionnaires [6,7].

Pain in the musician can generate disability to play an instrument [8], leading to a PRMD. In Spain, these problems are not recognized as an occupational disease [9], so it is important to know how many musicians have pain and disability and the relationship between both symptoms in order to adopt the appropriate measures in prevention and treatment and to provide information that can help the musician in the inclusion of their problems as an occupational disease. It is necessary to use culturally adapted and psychometrically validated instruments to study the presence of pain and disability, therefore the following objectives are proposed: (i) to evaluate the prevalence of musculoskeletal pain using the Spanish version of the Standardised Nordic Questionnaire (SNQ); (ii) to investigate the neck disability with the Neck Disability Index (NDI), the shoulder disability with the Shoulder and Pain Disability Index (SPADI), and the lower back disability with the Oswestry Disability Index (ODI); and (iii) to determine the associated factors between experiencing pain in the last seven days after playing an instrument and with the individual characteristics of the musician.

## 2. Materials and Methods

This study is a population-based, cross-sectional, descriptive study conducted from February, 2018 to December, 2018 (Appendix A). All procedures performed in studies involving human participants were in accordance with the ethical standards of academic board of the Doctoral Program in Health Science of University of Alcalá (D420/092015) and with the Declaration of Helsinki 1964 and its later amendments or comparable ethical standards. Written informed consent was obtained from all individuals.

### 2.1. Participants

An exponential discriminative snowball sampling technique has been used. An email containing the link to the “open questionnaire” (open for each visitor of a site) was sent to the Directors of public and private music schools, conservatories, and orchestras in the Community of Madrid (Spain). The link to the questionnaire was also administered through a Facebook (Facebook, Menlo Park, CA, USA) event and given to the Municipal Band of Alcobendas for its dissemination via WhatsApp. The inclusion criteria were musicians over 16 years old who played at least one musical instrument for a minimum of 5 h per week and who were native Spanish speakers to ensure understanding of the validated Spanish questionnaires.

### 2.2. Procedure

Before starting the questionnaires, the participants were given information about it, the available time to complete it, the main researcher, and the purpose of the study, and they gave their consent to participate.

The questionnaires were administered by means of the Google Forms platform, which allows for access and entry on any type of electronic device with an Internet connection. The use of self-completed online questionnaires to collect sample data in prevalence studies has been validated to be a useful method [7,10,11,12]. Furthermore, one of the questionnaires used, the SNQ, has been validated in an online format [13], proving to be a valid, reliable, and feasible tool. The description of the methodology was made following the Checklist for Reporting Results of Internet E-Surveys (CHERRIES) (Appendix A), a tool created to improve the description of the articles in which online questionnaires are used [14]. The Google Forms platform generates an Excel document with the answers. Before handing out the questionnaire, it was tested to verify its usability and technical functionality.

The online questionnaire included the SNQ, the ODI, the NDI, and the SPADI Spanish versions. It consisted of 16 sections of different length. In the first two sections, they were given information about the questionnaire and the purpose of the study and they were asked to give their consent to participate. In Section 3, sociodemographic data were asked (12 items). Section 4 included the general questionnaire of the SNQ. The remaining sections included the three specific questionnaires of the SNQ (each divided into three sections), the ODI, the NDI, the SPADI and a comments section. To speed up the questionnaire, some negative responses to questions from the specific questionnaires jumped to subsequent sections. Replying was a requirement in order to move on to the next section. The answers could only be sent if the questionnaire was fully completed. Participants could not review the answers at the end of the questionnaire. When sending the answers, the screen displayed a message thanking them for the participation. Duplicity of data was controlled with initials, gender and age. The response rate could not be controlled. The questionnaire collection time was 6 months (from April to October, 2018).

### 2.3. Questionnaires

The SNQ is a self-administered questionnaire. It consists of a general questionnaire with 27 items that assesses the presence of pain in nine body areas and the impact on the activities that cause the pain, and on three specific questionnaires to deepen the symptoms of the lumbar area (nine items), neck (nine items) and shoulders (10 items) [15].

The ODI is a self-applied questionnaire specific for low back pain that measures limitations in daily activities. The questionnaire consists of 10 items with a total score for each item of 5. The answers are added and multiplied by two to obtain a percentage. A higher score shows a higher level of disability [16].

The NDI was developed as a modification of the ODI. It is a self-completed questionnaire specific for cervical pain that measures limitations in daily activities. The score is obtained in the same way as in the ODI. A higher score shows a higher level of disability [17].

The SPADI is a self-administered questionnaire that consists of two dimensions: pain and functional activities. The pain dimension consists of five questions regarding the severity of pain. Functional activities are assessed with eight questions designed to measure the degree of difficulty with various activities of daily living that require upper-extremity use. The scoring percentage for each dimension and a total percentage of the questionnaire can be obtained. The score can be in a range between 0 (best pain or disability) and 100 (worst pain or disability) [18].

All of them were validated in Spanish [13,19,20,21].

### 2.4. Statistical Methods

To calculate the sample size, we used the results obtained by Leaver et al., in order to create a study that uses the SNQ and that analyses orchestral instruments in general, not just an instrumental group [22]. A sample size of 186 randomly selected subjects sufficed to estimate with a 95% confidence and a precision of +/- 5 percent units, a population percentage considered to be around 86%. A replacement rate of 0% was anticipated.

Sociodemographic data and characteristics of the instrumental practice were analysed using descriptive statistics with median as measure of central tendency, and the interquartile range (IQR) as measure of dispersion. Association of the sociodemographic data and the characteristics of the instrumental practice with the symptomatology in the last seven days was estimated using logistic regression. The variables with *p*-value < 0.250 were included in the logistic regression [23,24]. Variables with less than 4 subjects were not analysed because they were unbalanced. Prevalence odds ratios (OR) were calculated with a 95% confidence interval (CI) and the level of significance was *p* < 0.05. The data were analysed with the Statistical Package for the Social Sciences (SPSS) (IBM Corp. Armonk, NY, USA), version 24.0.

## 3. Results

### 3.1. Musicians’ Characteristics

A total of 361 responses were received. From these, 213 answers met the inclusion criteria. The sample consisted of 114 males (53.5%) and 99 females (46.5%) with a median age of 26 (IQR 18.5). The sociodemographic data and characteristics of the instrumental practice and its comparison by gender and symptoms in the last seven days can be seen in Table 1 and Table 2.

### 3.2. Prevalence and Disability of PRMD

The SNQ results show that 94.8% of musicians presented at least one symptomatic region in the last 12 months and 72.3% in the last seven days (Table 3). The most frequent areas of pain were the neck, shoulders, lower back and wrists/hands, for both periods. Pain in the neck, shoulders, low back and wrists/hands was more inhibiting in usual work for the musician. A total of 44.1% visited a health professional because of the neck, 31.0% because of the shoulders and 35.7% because of the lower back. Regarding the number of symptomatic areas in the last seven days, it was more frequent to experience symptoms in two to four regions than in only one area. Women had more affected areas than men measured with SNQ (*p* = 0.002) (Table 4).

More than half of the sample had a minimal disability or had no disability due to neck or lower back problems, measured with NDI and ODI, respectively. Pain in the shoulder measured with the SPADI pain subscale was worse (18 median (46 IQR)) than with the disability subscale (2.5 (18.75)). Women had greater neck and shoulder disability than men measured with NDI and SPADI (*p* < 0.001) (Table 5).

Among the musicians with pain, neck pain caused greater disability than pain in the lower back region (moderate and severe disability in neck: 15.5% and 3.1% respectively; moderate and severe disability in low back: 2.7% and 0% respectively). Musicians with shoulder pain had more pain than disability. Again, women had greater neck and shoulder disabilities than men measured with NDI and SPADI (*p* = 0.010 and *p* = 0.003 respectively) (Table 6).

### 3.3. Associated Factors of PRMD

The main factors associated with the presence of musculoskeletal pain in the last seven days were gender, BMI and weekly playing time. Women were more likely than men to report musculoskeletal pain (OR 4.38, CI 2.11−9.12). Musicians who were overweight and obese had a higher risk of presenting musculoskeletal pain (OR 5.32, CI 2.18−12.97). Musicians who played 15 h or more per week had a higher risk of reporting musculoskeletal pain (OR 3.86, CI 1.80−8.29) (Table 7).

When analyzing the associated factors according to the body area with the presence of pain in the last seven days, no association was found for the variables of age, dominant hand, years playing, elevated arm position while playing or asymmetric instrument. Regarding gender, an association is found between the female gender and the presence of pain in the neck, shoulders, elbows, wrists/hands, upper back, low back and ankles/feet, being greater for neck (OR 4.19, CI 2.19−8.03) and shoulders (OR 3.61, CI 1.95−6.69). There is a relationship between a higher BMI and pain in the neck, wrists/hands, low back, knees and ankles/feet, in this case being higher in the knees (OR 4.90, CI 2.01−11.91) and ankles/feet (OR 4.12, CI 1.34−12.66). Finally, an association is observed between playing more than 14 h and the presence of pain in the neck (OR 2.60, CI 1.38−4.88), shoulders (OR 2.03, CI 1.09−3.78) and wrists / hands (OR 1.91, 1.02−3.58) (Table 7 and Table 8).

## 4. Discussion

This is the first published study to address the prevalence of musculoskeletal pain among Spanish musicians using a validated Spanish version of the SNQ [13]. It also identifies associated factors with this prevalence and explores the disability produced by neck, shoulder and lower back pain.

The 94.8% and the 72.3% of musicians who participated in this study had experienced playing-related musculoskeletal pain within the past 12 months and the past seven days, respectively. This pain generates a disability from mild to severe depending on the affected regions. This is in accordance with other studies in which the prevalence of complaints in the last 12 months ranges between 62% and 93% according to the review carried out by Kok et al. [3,5]. Following this review, Kochem and Silva reported a 12-month prevalence of 86.8% and a seven-day prevalence of 72.3% in violinists [5]. Thus, musculoskeletal pain is shown as a major health problem among musicians. The sample recruitment method prevents us from knowing the response rate and probably encouraged musicians who have some musculoskeletal pain or problem that affects their instrumental practice to participate. In addition, in the data referring to the last 12 months, memory bias may occur. Despite this, the results are similar to those obtained in samples of musicians from other countries, as indicated above. Although this result cannot be generalized to the entire population of musicians, it does show clear evidence that there is a large number of musicians with musculoskeletal pain, and that the inclusion of musculoskeletal problems in musicians should therefore be considered within the legislation of occupational diseases in Spain.

Regarding the areas, Kok et al. concluded that the most affected areas are the neck and shoulders and that the least affected areas are the elbows, which is in accordance with our results [3]. Our study also finds a high prevalence of lower back pain according to Sousa et al. and Rodríguez-Romero et al. [6,25].

In the current study, it was found that musicians who play string instruments are more likely to present musculoskeletal pain in the last seven days, although these results are not statistically significant (*p* = 0.141). Kok et al. found that no specific instrument group had an evidently higher prevalence rate of musculoskeletal complaints. However, brass instrumentalists were reported to have the lowest prevalence rates of musculoskeletal complaints [3,10].

More than 31% of the musicians responded that they visited a health professional regarding musculoskeletal problems in the neck, shoulders or lower back. This percentage is similar to the study conducted by Ioannou and Altenmüller [26], in which 35.2% of affected musicians visited a health professional, and exceeds the 10.3% obtained by Kochem and Silva [5]. Ioannou and Altenmüller’s study sample is made up of students with a daily practice time of 4.4 h (± 1.14) [26], while Kochem and Silva’s sample is composed by professional musicians, who play 23.3 h weekly (± 12.2) [5]. From these data, we can assume that students make a greater effort to practice to pass their exams, and they need to be well to perform correctly, which would lead to a greater need to reach out to health professionals.

Regarding disability, in our study, women presented greater neck and shoulder pain and disability than men measured with the NDI and SPADI. Few studies assess disability in musicians. Rodríguez-Romero et al. used the NDI and the questionnaire “Disabilities of the Arm, Shoulder and Hand” (DASH) to analyse this disability, obtaining similar results to ours for the NDI [6]. Kochem and Silva used the DASH questionnaire and concluded that violinists who scored >10.1 points were more likely to develop PRMD in the last 12 months (OR 3.7, CI 1.6−8.6) and in the last seven days (OR 3.6, CI 1.1−11.3) [5]. In our study, we decided to use the NDI and the SPADI after analysing in the literature review that neck and shoulders are the two areas with the highest prevalence of complaints by the musician [3]. In addition, it was decided that the ODI be used, because it was an area that also presented a high prevalence [6,25,27], but its disability had not been studied before.

Furthermore, we also found an association between the female gender and the presence of pain in the last seven days (2.39 (1.29−4.44)). Females have twice the risk of suffering musculoskeletal pain than men. Paarup et al. reflected similar results (OR 3.0, CI 1.9−4.5) [4]. Moreover, women showed musculoskeletal pain in the most anatomical regions. Several authors stated the relationship between women and pain in the neck [4,5,6,27] and there is a tendency to observe a relationship with complaints in the upper limbs and dorsal area [4,5,27]. Rodríguez-Romero et al. also found an association with the lumbar area (OR 3.9, CI 1.7−8.7) [6].

The preponderance of women reporting pain in more regions and more disability than men is in accordance with findings investigating risk factors for pain and sex differences in body composition [28,29]. Anthropometric features, such as greater strength and better aerobic capacity in males than females [30], along with differences in the recruitment of the muscles that stabilize the scapula during glenohumeral abduction, affecting mainly females, and a higher prevalence of joint hypermobility in women because of hormonal aspects could explain the higher prevalence ratios for women [31,32].

Our results show a relationship between being overweight and obesity and the presence of symptoms in the last seven days (OR 5.32, CI 2.18−12.97). Zaza and Farewell found this same association, but in a lower regard than in the present study (OR 1.187, CI 1.045−1.348) [33]. However, Kochem and Silva contradict these results, affirming that musicians with a BMI < 25 kg/m^2^ are more likely to experience symptoms in the left hand (OR 1.88, CI 1.03−3.46) [5]. They believe that a possible explanation may be the relationship between a lower BMI and a lower muscular tropism. The upper limb of the violinist has a more static action supporting the instrument’s weight, and thus, a low BMI may result in a lower muscle endurance interfering in the artistic performance [5]. However, in our study, the relationship found is the opposite. Traditionally, the relationship between obesity and pain has simply been regarded as resulting from the intermediary effect of arthritis due to increased joint loading [34]. Currently, the adipose tissue is considered to be an endocrine organ, promoting low-grade systemic inflammation by secreting adipokines [35]. Obesity has been associated with markers of chronic inflammation, such as levels of C-reactive protein, tumour necrosis factor α, amyloid A and interleukin−6, as well as white blood cell counts [35,36,37]. Thus, the impact of obesity on various musculoskeletal conditions may stem not only from the biomechanical stress of obesity, but also from systemic effects.

Rodríguez-Romero et al. found a relationship between playing more than 33 h weekly and the presence of shoulder pain in the last seven days [6]. In our study, we also observed this relationship, in addition to the neck and wrists/hands. Muscle pain is the most common symptom of musculoskeletal disorders [38]. A model proposed by Johansson and Sojka in 1991 implied that sustained muscle contractions, inflammation, and/or ischaemia could start a “vicious circle” of muscle stiffness to primary and secondary muscles and thereby preserve or increase the production of metabolites and the high activity in the chemosensitive nerve endings [39].

In our study, there is no relationship between playing with an elevated arm position and the presence of symptoms in the last seven days. However, the results of Nyman et al. showed that playing with an elevated arm position is a risk factor for musculoskeletal problems in the neck and shoulders, especially if it is for more than three hours a day (OR 5.35, CI 1.96−14.62) [40]. This difference may be due to how the presence or absence of pain is assessed in the Nyman et al. questionnaire. In their study, they use a questionnaire to ask about pain in the neck, shoulders and between the shoulders blades with a five-graded response scale for each question (0, completely healthy; 1, a little pain, but no problem; 2, quite a bit of pain, but it is possible to play; 3, very much pain, have to avoid certain movements; and 4, so much pain that I sometimes cannot work). A subject was considered as having neck-shoulder complaints if he/she had scored at least a “2” on at least one of the three items [40]. While they use a graduated scale, selecting the cases that score at least a “2”, in the present study, a dichotomous question is used, so that we obtain a greater number of subjects with complaints, and therefore the elevated position of the arm cannot be considered a risk factor.

### Strengths and Limitations

To our knowledge, this is the first study conducted in Spain with a questionnaire that is cross-culturally adapted and validated to Spanish musicians. This is also the first study that assesses the disability of the lower back, allowing us to understand this region more deeply, with a high prevalence of musculoskeletal complaints.

Some limitations should be considered when interpreting the results of prevalence of pain and disability. The response rate could not be controlled. Even so, studies of prevalence in musicians usually have a low response rate [3]. The sampling method used is not probabilistic, so the representativeness of the sample cannot be assured. To reduce this bias, an email was sent to all music schools and conservatories in Madrid. Despite the sampling technique used, the results of this study are like other studies conducted with other sampling techniques, especially regarding the complaints from the last seven days. For complaints from the last 12 months, higher results were obtained. A possible explanation is that musicians with some complaints were more motivated to answer the questionnaires. Another explanation is the influence of recall bias. As musicians answered the questionnaire at different times of the year, they may be more affected during concert times than during their holidays. The SNQ is a good screening tool, because it shows good agreement with the functional clinical assessment, but it should not be used as a tool to confirm the diagnosis of a disorder or pathology due to a significant number of false positives [41]. The questionnaire did not inquire about musculoskeletal pain in the face and the temporomandibular joint because this region was not included in the questionnaire used. It would also be important to know the issues of this region in musicians.

## 5. Conclusions

In summary, Spanish musicians show a high prevalence of musculoskeletal pain, located mainly in the neck, shoulders, lower back and wrists/hands. There are more women with mild to severe disability in the neck than men. Women have higher levels of disability and pain than men in the shoulders. There is a clear relationship between the presence of these complaints and the female gender, being overweight, obesity and playing more than 14 h a week. This study highlights the need to plan and adapt preventive strategies to prevent occupational disabilities among musicians. Further studies are needed to analyse the prevalence of pain in the musician using other sampling methods.

## Figures and Tables

**Table 1 ijerph-17-03991-t001:** Sociodemographic and instrumental practice data compared by gender (*n* = 213).

Variable	Total	Females (*n* = 99)	Males (*n* = 114)	*p*-Value
Age (years) (Median (IQR)) *	26 (18.5)	23 (14)	29.5 (19.25)	<0.001
BMI (*n* (%))				0.002
Underweight and Adequate Weight ( <25 kg/m^2^)	150 (70.4)	80 (80.8)	70 (61.4)	
Overweight and Obesity ( ≥25 kg/m^2^)	63 (29.6)	19 (19.2)	44 (38.6)	
Dominant Hand (*n* (%))				0.988
Right Handed	198 (93.0)	92 (92.9)	106 (93.0)	
Left Handed	15 (7.0)	7 (7.1)	8 (7.0)	
Years Playing (Median (IQR)) *	15 (14)	14 (11)	18 (18)	0.036
Weekly Playing Time (*n* (%))				0.003
<14 h	128 (60.1)	70 (70.7)	58 (50.9)	
≥15 h	85 (39.9)	29 (29.3)	56 (49.1)	
Instrumental Group (*n* (%))				<0.001
String	39 (19.4)	31 (33.0)	8 (7.5)	
Keyboard	34 (16.9)	23 (24.5)	11 (10.3)	
Plucked	29 (14.4)	4 (4.3)	25 (23.4)	
Woodwind	60 (29.9)	25 (26.6)	35 (32.7)	
Brass	20 (10.0)	6 (6.4)	14 (13.1)	
Percussion	19 (9.5)	5 (5.3)	14 (13.1)	
Number of Instruments Played (*n* (%))				0.403
1	199 (93.4)	94 (94.9)	105 (92.1)	
>1	14 (6.6)	5 (5.1)	9 (7.9)	
Elevated Arm Position while Playing (*n* (%))				0.008
Yes	53 (24.9)	33 (33.3)	20 (17.5)	
No	160 (75.1)	66 (66.7)	94 (82.5)	
Asymmetric Instrument (*n* (%))				0.957
Yes	105 (49.3)	49 (49.5)	56 (49.1)	
No	108 (50.7)	50 (50.5)	58 (50.9)	
Symptomatology in the last 7 days (*n* (%))				0.010
Yes	154 (72.3)	80 (80.8)	74 (64.9)	
No	59 (27.7)	19 (19.2)	40 (35.1)	

* Calculated with Mann–Whitney U test; No normal distribution: Median (IQR): median (interquartile range); BMI: Body mass index.

**Table 2 ijerph-17-03991-t002:** Sociodemographic and instrumental practice data compared by symptomatology in the last seven days (*n* = 213).

Variable	Total	Symptomatic (*n* = 154)	Asymptomatic (*n* = 59)	*p*-Value
Gender (*n* (%))				0.010
Females	99 (46.5)	80 (51.9)	19 (32.2)	
Males	114 (53.5)	74 (48.1)	40 (67.8)	
Age (years) (Median (IQR)) *	26 (18.5)	26.5 (18.25)	26 (22)	0.658
BMI (*n* (%))				0.002
Underweight and Adequate Weight (<25 kg/m^2^)	150 (70.4)	99 (64.3)	51 (86.4)	
Overweight and Obesity (≥ 25 kg/m^2^)	63 (29.6)	55 (35.7)	8 (13.6)	
Dominant Hand (*n* (%))				0.926
Right Handed	198 (93.0)	143 (92.9)	55 (93.2)	
Left Handed	15 (7.0)	11 (7.1)	4 (6.8)	
Years Playing (Median (IQR)) *	15 (14)	16 (13.5)	15 (15)	0.513
Weekly Playing Time (*n* (%))				0.003
<14 h	128 (60.1)	83 (53.9)	45 (76.3)	
≥15 h	85 (39.9)	71 (46.1)	14 (23.7)	
Instrumental Group (*n* (%))				0.141
String	39 (19.4)	34 (23.6)	5 (8.8)	
Keyboard	34 (16.9)	21 (14.6)	13 (22.8)	
Plucked	29 (14.4)	20 (13.9)	9 (15.8)	
Woodwind	60 (29.9)	39 (27.1)	21 (36.8)	
Brass	20 (10.0)	15 (10.4)	5 (8.8)	
Percussion	19 (9.5)	15 (10.4)	4 (7.0)	
Number of Instruments Played (*n* (%))				0.246
1	199 (93.4)	142 (92.2)	57 (96.6)	
>1	14 (6.6)	12 (7.8)	2 (3.4)	
Elevated Arm Position while Playing (*n* (%))				0.097
Yes	53 (24.9)	43 (27.9)	10 (16.9)	
No	160 (75.1)	111 (72.1)	49 (83.1)	
Asymmetric Instrument (*n* (%))				0.211
Yes	105 (49.3)	80 (51.9)	25 (42.4)	
No	108 (50.7)	74 (48.1)	34 (57.6)	

* Calculated with Mann–Whitney U test; No normal distribution: Median (IQR): median (interquartile range); BMI: Body mass index.

**Table 3 ijerph-17-03991-t003:** Number of complaints by region and number of affected areas (*n* = 213).

Location of Trouble	Questions					
	Have You at Any Time during the Last 12 Months Had Trouble (Ache, Pain, Discomfort) in? (*n* (%))		Have You at Any Time during the Last 12 Months Been Prevented from Doing Your Normal Work (at Home or away from Home) Because of the Trouble? (*n* (%))		Have You Had Trouble at Any Time during the Last Seven Days? (*n* (%))	
	Yes	No	Yes	No	Yes	No
Any Area	202 (94.8)	11 (5.2)	99 (46.5)	114 (53.5)	154 (72.3)	59 (27.7)
Neck	157 (73.7)	56 (26.3)	41 (19.2)	172 (80.8)	97 (45.5)	116 (54.5)
Shoulders	126 (59.2)	87 (40.8)	33 (15.5)	180 (84.5)	79 (31.7)	134 (62.9)
Elbows	37 (17.4)	176 (82.6)	13 (6.1)	200 (93.9)	18 (8.5)	195 (91.5)
Wrists/Hands	110 (51.6)	103 (48.4)	42 (19.7)	171 (80.3)	68 (31.9)	145 (68.1)
Upper Back	93 (43.7)	120 (56.3)	25 (11.7)	188 (88.3)	58 (27.2)	155 (72.8)
Low Back	122 (57.3)	91 (42.7)	40 (18.8)	173 (81.2)	73 (34.3)	140 (65.7)
Hips/Thighs	31 (14.6)	182 (85.4)	12 (5.6)	201 (94.4)	12 (5.6)	201 (94.4)
Knees	45 (21.1)	168 (78.9)	9 (4.2)	204 (95.8)	24 (11.3)	189 (88.7)
Ankles/Feet	30 (14.1)	183 (85.9)	6 (2.8)	207 (97.2)	15 (7.0)	198 (93.0)

**Table 4 ijerph-17-03991-t004:** Number of affected areas in the last seven days and comparison by gender (*n* = 213).

Affected Areas	Total	Women	Men	*p*-Value
No Problem	59 (27.7)	19 (19.2)	40 (35.1)	0.002
1 Area	30 (14.1)	11 (11.1)	19 (16.1)	
2−4 Areas	102 (47.9)	52 (52.2)	50 (43.9)	
5 or more Areas	22 (10.3)	17 (17.2)	5 (4.4)	

**Table 5 ijerph-17-03991-t005:** Disability of participants and comparison by gender (*n* = 213).

Index	Pain/Disability			
	Total	Women	Men	*p*-Value
NDI (*n* (%))				<0.001
No Disability (0%−8%)	122 (57.3)	40 (40.4)	82 (71.9)	
Mild Disability (10%−28%)	72 (33.8)	44 (44.4)	28 (24.6)	
Moderate Disability (30%−48%)	16 (7.5)	12 (12.1)	4 (3.5)	
Severe Disability (50%−64%)	3 (1.4)	3 (3.0)	0 (0.0)	
Complete Disability	0 (0.0%)	0 (0.0)	0 (0.0)	
ODI (*n* (%))				0.303
Minimal Disability (0%−20%)	189 (88.7)	85 (85.9)	104 (91.2)	
Moderate Disability (21%−40%)	21 (9.9)	13 (13.1)	8 (7.0)	
Severe Disability (41%−60%)	3 (1.4)	1 (1.0)	2 (1.8)	
Crippled (61%−80%)	0 (0.0)	0 (0.0)	0 (0.0)	
Bed-Bound (81%−100%)	0 (0.0)	0 (0.0)	0 (0.0)	
SPADI (*Median* (*IQR*))				
Pain	18 (46)	32 (50)	10 (26)	<0.001 *
Disability	2.5 (18.75)	8.75 (23.75)	0 (9.06)	<0.001 *
Total Score	9.23 (27.31)	19.23 (31.54)	4.23 (18.27)	<0.001 *

* Calculated with Mann–Whitney U test; No normal distribution: Median (IQR): median (interquartile range); NDI: Neck Disability Index; ODI: Oswestry Disability Index; SPADI: Shoulder Pain and Disability Index.

**Table 6 ijerph-17-03991-t006:** Disability of symptomatic participants and comparison by gender.

Index	Pain/Disability			
	Total	Women	Men	*p*-Value
NDI (*n* = 97) (*n* (%))				0.010
No Disability (0%−8%)	32 (33.0)	12 (21.1)	20 (50)	
Mild Disability (10%−28%)	47 (48.5)	30 (52.6)	17 (42.5)	
Moderate Disability (30%−48%)	15 (15.5)	12 (21.1)	3 (7.5)	
Severe Disability (50%−64%)	3 (3.1)	3 (5.3)	0 (0.0)	
Complete Disability	0 (0.0)	0 (0.0)	0 (0.0)	
ODI (*n* = 73) (*n* (%))				0.965
Minimal Disability (0%−20%)	59 (80.0)	35 (81.4)	24 (80.0)	
Moderate Disability (21%−40%)	12 (16.4)	7 (16.3)	5 (16.7)	
Severe Disability (41%−60%)	2 (2.7)	1 (2.3)	1 (3.3)	
Crippled (61%−80%)	0 (0.0)	0 (0.0)	0 (0.0)	
Bed-Bound (81%−100%)	0 (0.0)	0 (0.0)	0 (0.0)	
SPADI (*n* = 79) (*Median* (*IQR*))				
Pain	38 (44)	46 (37.5)	20 (40)	<0.001 *
Disability	12.5 (25)	15 (25)	3.75 (21.87)	0.017 *
Total Score	23.85 (30)	28.08 (24.81)	10 (28.07)	0.003 *

* Calculated with Mann–Whitney U test; No normal distribution: Median (IQR): median (interquartile range); NDI: Neck Disability Index; ODI: Oswestry Disability Index; SPADI: Shoulder Pain and Disability Index.

**Table 7 ijerph-17-03991-t007:** Factors associated with pain in any area, neck, and upper limbs.

Variable	Any Area	Neck	Shoulders	Elbows	Wrists/Hands
Gender					
Males	1	1	1	1	1
Females	4.38 (2.11−9.12) *	4.19 (2.19−8.03) *	3.61 (1.95−6.69) *	2.48 (0.89−6.88)	2.59 (1.36−4.93) *
Age (Years)	NC	NC	0.99 (0.97−1.01)	NC	NC
BMI			NC	NC	
Underweight and Adequate Weight	1	1			1
Overweight and Obesity	5.32 (2.18−12.97) *	2.85 (1.45−5.63) *			1.98 (1.01−3.86) *
Dominant Hand	NC	NC	NC	NC	
Right Handed					1
Left Handed					0.14 (0.02−1.07)
Years Playing	NC	NC	NC	1.03 (0.99−1.07)	NC
Weekly Time Playing				NC	11.91 (1.02−3.58) *
<14 h	1	1	1		
≥15 h	3.86 (1.80−8.29) *	2.60 (1.38−4.88) *	2.03 (1.09−3.78) *		
Elevated Arm Position while Playing		NC	NC	NC	NC
Yes	1.52 (0.62−3.70)				
No	1				
Asymmetric Instrument					
Yes	1.82 (0.93−3.55)	1.77 (0.98−3.19)	1.73 (0.96−3.11)	0.49 (0.17−1.38)	0.70 (0.38−1.29)
No	1	1	1	1	1

Odds ratio (95% CI); * *p* < 0,05; NC: Not calculated.

**Table 8 ijerph-17-03991-t008:** Factors associated with pain in back and lower limbs.

Variable	Upper Back	Low Back	Hips/Thighs	Knees	Ankles/Feet
Gender				NC	
Males	1	1	1		1
Females	2.39 (1.29−4.44) *	2.76 (1.48−5.14) *	3.25 (0.89−11.80)		3.50 (1.08−11.33) *
Age (Years)	0.98 (0.95−1.01)	NC	NC	NC	NC
BMI	NC				
Underweight and Adequate Weight		1	1	1	1
Overweight and Obesity		2.74 (1.42−5.30) *	3.40 (0.99−11.61)	4.90 (2.01−11.91) *	4.12 (1.34−12.66) *
Dominant Hand		NC	NC	NC	NC
Right handed	1				
Left Handed	1.84 (0.60−5.68)				
Years Playing	NC	NC	NC	NC	NC
Weekly Time Playing	NC	NC	NC	NC	NC
<14 h					
≥15 h					
Elevated Arm Position While Playing		NC	NC	NC	NC
Yes	0.98 (0.45−2.14)				
No	1				
Asymmetric Instrument		NC	NC	NC	NC
Yes	1.52 (0.82−2.84)				
No	1				

Odds ratio (95% CI); * *p* < 0.05; NC: Not calculated.

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
