# Peer review of "Prevalence, Disability and Associated Factors of Playing-Related Musculoskeletal Pain among Musicians: A Population-Based Cross-Sectional Descriptive Study"

_ijerph, 2020, doi:10.3390/ijerph17113991_

Round 1
Reviewer 1 Report
General comments:
The authors have written an interesting paper reporting on the prevalence of musculoskeletal complaints in musicians, as well as risk factors for these complaints. The method of recruiting participants gives a large risk of sampling bias, which should be addressed more clearly in the paper. Furthermore, the paper would be more informative if multiple logistic regression was reported instead of binary regression, and ethical approval and table 6 were explained more clearly. This suggests that changes should be made to the Result and Discussion part. I also think the conclusion would need rephrasing.
More detailed comments:
Abstract
The study has a cross-sectional design and therefore the musicians are not “prospectively” selected. Data are rather “retrospective”.
Introduction
page 2 line 54-
It is important to detect the musician with pain before developing a PRMD. In addition, it is relevant to know the disability caused by this pain. Why are these things important? Please explain and add relevant references.
Materials and methods
page 2 line 66
"CHERRIES (8)" Is this a relevant reference or should it be another reference? It does not refer to the original study about Cherries.
Page 2
Participants
What do you mean by the “open questionnaire”?
Please give more information about inclusion criteria
Please give more information about exclusion criteria
What was the time-period of gathering the questionnaires?
Page 3
Questionnaires
Please for the questionnaires provide more information about measurement properties and number of questions and min – max score.
Ethics
Which ethical committee/authority approved the study? Approval number and year?
Statistical methods
It is not reported which variables were chosen for the regression analysis and what was the rationale for these?
Table 2
Final variable in the Table “Symptomatology in the lost 7 days” is redundant.
Table 6
Please explain what you mean by Table 6. What is the difference with Table 5? It is unclear to me. What do you mean by “symptomatology” in the heading?
Table 7
What do you mean by NE “not evaluated”? Do you mean “not calculated” or do you mean that the calculations did not show “statistically significant difference”?
Table 7
It would be more interesting if you report results of multiple logistic regression (or both).
Page 10
Discussion
I think after reporting the results for “Prevalence” you should directly give a strong warning that there might be a high probability of sampling bias due to the way of recruiting participants. You should also discuss how this can influence your results.
Strength and limitations
Here you should also discuss the high risk of sampling bias. Even if you do not know the response rate, you might know the number of musicians studying at the institutions to which the questionnaire was sent? Another solution would be to exclude the part about overall “prevalence” and only report the more certain data about those musicians who actually (claimed they) had musculoskeletal complaints.
Conclusions
Due to results of multiple regression analyses it might be necessary to change the conclusion. Also, the high risk of sampling bias should be mentioned in the conclusion.
Author Response
Thank you for the relevant comments to the manuscript. The suggestions will help to improve the understanding and the quality of the manuscript. We have done our best to make the required improvements, in the text of the manuscript the amendments appear with track changes. The order and the list of references have changed because those underlined in yellow have been added to the text.
Please, see our reply (answers are also available in the attachment).
Point 1. Abstract
The study has a cross-sectional design and therefore the musicians are not “prospectively” selected. Data are rather “retrospective”.
Response 1: Thank you for the suggestion. We have decided to remove "prospectively" (page 1, line 20).
Point 2. Introduction, page 2 line 54
It is important to detect the musician with pain before developing a PRMD. In addition, it is relevant to know the disability caused by this pain. Why are these things important? Please explain and add relevant references.
Response 2: We have tried to give a better explanation to the importance of detecting the musician with pain and its relationship with disability. We hope that the importance of this study in Spanish population is now better understood (page 2, line 57-61).
Point 3. Materials and methods, page 2 line 66
"CHERRIES (8)" Is this a relevant reference or should it be another reference? It does not refer to the original study about Cherries.
Response 3: Thanks for the observation, but we consider the reference is correct. The reference has been checked, and it is the original article where the usefulness of the CHERRIES is explained to better describe the articles in which online questionnaires are used (page 3, line 99).
Point 4. Page 2, Participants
What do you mean by the “open questionnaire”?
Response 4: As the CHERRIES describes: An open survey is a survey open for each visitor of a site, while a closed survey is only open to a sample which the investigator knows (password-protected survey).
It has been clarified in the text of the article (page 2, line 82). Thank you.
Point 5. Please give more information about inclusion criteria. Please give more information about exclusion criteria
Response 5: The inclusion and exclusion criteria were those listed in the manuscript. An explanation has been added to why only native Spanish speakers were included (page 2, line 85-87). Thank you.
Point 6. What was the time-period of gathering the questionnaires?
Response 6: The collection period of the questionnaires has been added to the text (page 3, line 114).
Point 7. Page 3, Questionnaires
Please for the questionnaires provide more information about measurement properties and number of questions and min – max score.
Response 7: Thank you for the suggestion. This information has been included (page 3, line 115-134).
Point 8. Ethics
Which ethical committee/authority approved the study? Approval number and year?
Response 8: This information has been included (page 2, line 73-77). Thanks for pointing it out.
Point 9. Statistical methods
It is not reported which variables were chosen for the regression analysis and what was the rationale for these?
Response 9: The variables chosen for the regression analysis are specified in the text (“The variables with p-value < 0.250 have been included in the logistic regression (23, 24). Variables with less than 4 subjects were not analyzed because they were unbalanced.”). Bibliographic references that justify the choice of these variables have been included (page 4, line 150).
It was not possible to specify which variables have been used for each body area in this section, as they are different according to the region. These variables are shown in Table 7, specifying those not calculated as NC (Not Calculated).
Point 10. Table 2
Final variable in the Table “Symptomatology in the lost 7 days” is redundant.
Response 10: Thank you for the correction. The final variable in Table 2 has been removed. The first variable in Table 1 has also been removed for the same reason. Table 1 and Table 2 were originally a single table but they were modified to keep the vertical format.
Point 11. Table 6
Please explain what you mean by Table 6. What is the difference with Table 5? It is unclear to me. What do you mean by “symptomatology” in the heading?
Response 11: The aims were, on the one hand, what was the disability of the entire sample collected and, on the other hand, what was the disability of the musicians who had pain in the last 7 days (we refer to them as “symptomatic”). Table 5 shows the answer to the first aim and Table 6 to the second.
The title of Table 6 has been modified for a better understanding (page 8, line 204). In addition, it has been clarified in the paragraph prior to Table 6 (page 10, line 197-202).
Point 12. Table 7
What do you mean by NE “not evaluated”? Do you mean “not calculated” or do you mean that the calculations did not show “statistically significant difference”?
Response 12: NE (not evaluated) refers to variables that were not calculated. The p-value for each variable and body area was calculated and only those variables with p-value < 0.250 were included in the logistic regression model, and variables with less than 4 subjects were discarded because they were considered unbalanced. So, to make it better understood, in table 7 NE (Not Evaluated) has been rewritten by NC (Not Calculated).
Point 13. Table 7
It would be more interesting if you report results of multiple logistic regression (or both).
Response 13: In the present study, we evaluated the presence or absence of pain as a dependent variable, that is, a dichotomous variable (the outcome is binary). Therefore, a binary logistic regression (presence/absence of pain) is required. In this way, the regression model used in the study responds to the stated objectives.
References:
- Berger, D.E. Introduction to Binary Logistic Regression and Propensity Score Analysis. Working Paper 2017.
- Doménech Masson, J.M. Análisis multivariante: modelos de regresión logística; Gráficas SIGNO SA: Barcelona, España, 1999.
Point 14. Page 10, Discussion
I think after reporting the results for “Prevalence” you should directly give a strong warning that there might be a high probability of sampling bias due to the way of recruiting participants. You should also discuss how this can influence your results.
Response 14: The biases of the results on prevalence have been discussed and it has been explained what should be considered about them. I hope the importance of results is now better understood despite existing biases (page 11-12, line 248-256).
Point 15. Strength and limitations
Here you should also discuss the high risk of sampling bias. Even if you do not know the response rate, you might know the number of musicians studying at the institutions to which the questionnaire was sent? Another solution would be to exclude the part about overall “prevalence” and only report the more certain data about those musicians who actually (claimed they) had musculoskeletal complaints.
Response 15: It is not possible to know the population of musicians from music schools and bands in Madrid because many musicians belong to multiple groups. It has been decided to modify the text of the discussion to emphasize that the prevalence is not so important, but rather that there is a very large number of musicians with pain that must be taken into account in the law on occupational disease in Spain and that it is very important to propose health promotion and prevention programs in this occupational group.
Point 16. Conclusions
Due to results of multiple regression analyses it might be necessary to change the conclusion. Also, the high risk of sampling bias should be mentioned in the conclusion.
Response 16: Thank you. The conclusions have been rewritten, considering the biases of the study (page 14, line 378-379).

Reviewer 2 Report
The aims of this study were to assess the prevalence of musculoskeletal pain, to evaluate neck, shoulder and lower back disability, and to determine the associated factors with the presence of musculoskeletal pain among musicians. In general, this topic is important and worthy of investigation. The manuscript is also well-written, but I think it seems too fragmental in writing. Some paragraphs one or two sentences included can be merged together or reorganized properly. This should be checked and revised throughout the manuscript. The authors should also check the purposes of the questionnaires and indices employed in the study, the results can be therefore made, for example, disability, disorders, symptoms, pain, etc. I do not mean to say that the descriptions or term usages were wrong, just a reminder because of different questionnaire and indices were used in the study. In addition, the questionnaires and the indices were responded online and the validity issue should be clarified and convinced. Some specific comments for the manuscript are also as follows.
- Abstract, if the structured abstract is acceptable for the journal, please delete the underlines of the sections.
- L18, please check “disability” or “symptoms” were examined in the study.
- L23, Ninety four-point eight percent, can be briefly presented as 94.8%.
- L27, Musculoskeletal disorders are highly prevalent in musicians, disorders may be replaced by disorders symptoms. Please check it.
- L27, The descriptions in Conclusions were duplicated and need rewritten.
- L41, Playing-related 12-month prevalence ranged between 41 % and 93 %. Incomplete sentence; prevalence in what?
- L48-53, I suggest these two paragraphs can be merged to one paragraph.
- L63-66, merge these two paragraphs to one, and please give the ethical code.
- L67, I suggest that subsection 2.1 can be divided into two subsections for Participants and Procedure individually.
- L71, please give the specific date or duration.
- L81, People always question the quality and validity regarding the online questionnaire. The authors should convince the readers this issue, besides references of 7, 9-11 were cited. It needs more detail.
- L105-108, Move these statements into L63-66 and merge as one paragraph.
- L124-126, this paragraph conveyors very little information and can be removed.
- L129, how to confirm no prescreening effect happened since the return rate was relatively low (59%); or highlight it in the study limitation? (but the authors seem to accept the prescreening effect).
- L131, table 1 and table 2, should be Table 1 and Table 2, or Tables 1 and 2. Please check the wording throughout the manuscript.
- L186, some results may be more significant than the minimum level of 0.05, for example, BMI factor for any area and knees, gender factor for any area and neck. It needs more detail.
- L203-214, Please consider to merge these three paragraphs into the previous one.
- L236, similar results
- L271, lacking cited number of Nyman et al’s study.
- L283, please merge these paragraphs properly.
- L298, “A possible explanation is that musicians with some complaints have been more motivated to answer the questionnaires. Another explanation is the influence of recall bias.” This is a big problem and may make these results invalid. More reasonable explanations are needed.
- L312-313, merge into the previous one.
- References, please give the DOI.
Author Response
Thank you for the relevant comments to the manuscript. The suggestions will help to improve the understanding and the quality of the manuscript. We have done our best to make the required improvements, in the text of the manuscript the amendments appear with track changes. The order and the list of references have changed because those underlined in yellow have been added to the text.
Please, see our reply (answers are also available in the attachment).
Point 1. Abstract, if the structured abstract is acceptable for the journal, please delete the underlines of the sections.
Response 1: Thanks for the suggestion. The underlines of the sections have been deleted.
Point 2. L18, please check “disability” or “symptoms” were examined in the study.
Response 2: "Disability" and "symptoms" have been checked. Thank you.
Point 3. L23, Ninety four-point eight percent, can be briefly presented as 94.8%.
Response 3: Thank you. It has been replaced (page 1, line 23).
Point 4. L27, Musculoskeletal disorders are highly prevalent in musicians, disorders may be replaced by disorders symptoms. Please check it.
Response 4: Thank you for the correction. It has been replaced by musculoskeletal disorders symptoms (page 1, line 30-31).
Point 5. L27, The descriptions in Conclusions were duplicated and need rewritten.
Response 5: The results have been described in another way so as not to duplicate the information in the "conclusions" section (page 1, line 25).
Point 6. L41, Playing-related 12-month prevalence ranged between 41 % and 93 %. Incomplete sentence; prevalence in what?
Response 6: The sentence has been completed for better understanding (page 2, line 45-46).
Point 7. L48-53, I suggest these two paragraphs can be merged to one paragraph.
Response 7: Thank you for the suggestion. Now both sentences are written in the same paragraph (page 2, line 53).
Point 8. L63-66, merge these two paragraphs to one, and please give the ethical code.
Response 8: Thank you for the suggestion. Now both sentences are written in the same paragraph.
Regarding the ethical code, this information has been included (page 2, line 73-77). Thanks for pointing it out.
Point 9. L67, I suggest that subsection 2.1 can be divided into two subsections for Participants and Procedure individually.
Response 9: Section 2.1 has been divided into two sections: "2.1. Participants" and "2.2. Procedure" (page 2, line 80, 88).
Point 10. L71, please give the specific date or duration.
Response 10: The data collection period has been included in the text (page 3, line 114).
Point 11. L81, People always question the quality and validity regarding the online questionnaire. The authors should convince the readers this issue, besides references of 7, 9-11 were cited. It needs more detail.
Response 11: Online questionnaires are increasingly used today as it is an accessible method of reaching the population. In addition, the Standardised Nordic Questionnaire has been validated in online format, proving to be valid, reliable, and feasible. It was also decided to describe the methodology according to CHERRIES to improve the validity of the study, as it is indicated on page 3, line 95-99. This paragraph has been rewritten to show this data.
Point 12. L105-108, Move these statements into L63-66 and merge as one paragraph.
Response 12: Thank you for the suggestion, we have transferred this paragraph to the indicated place (page 2, line 73-77).
Point 13. L124-126, this paragraph conveyors very little information and can be removed.
Response 13: Sorry for the confusion. This paragraph, that belonged to the journal template, has been removed (page 4, line 155-157).
Point 14. L129, how to confirm no prescreening effect happened since the return rate was relatively low (59%); or highlight it in the study limitation? (but the authors seem to accept the prescreening effect).
Response 14: In the discussion, the impossibility of knowing the response rate and the limitations it implies for this study have been explained.
Point 15. L131, table 1 and table 2, should be Table 1 and Table 2, or Tables 1 and 2. Please check the wording throughout the manuscript.
Response 15: Thank you. Fixed in all text "table" by "Table"
Point 16. L186, some results may be more significant than the minimum level of 0.05, for example, BMI factor for any area and knees, gender factor for any area and neck. It needs more detail.
Response 16: Thank you for the suggestion. The relationship between the presence of pain in the different body areas and the variables studied has been described (page 10, line 219-228).
Point 17. L203-214, Please consider to merge these three paragraphs into the previous one.
Response 17: The first paragraph has been joined to the previous paragraph. The other two paragraphs have been decided to keep them as independent paragraphs because they refer to different questions: "pain prevalence in each body area" and "pain prevalence according to the instrument". (page 11, line 247).
Point 18. L236, similar results
Response 18: Thank you for the suggestion. "The same" has been replaced by "similar" (page 12, line 288).
Point 19. L271, lacking cited number of Nyman et al’s study.
Response 19: Nyman's reference is reference 40, located at the end of the paragraph (page 13, line 332). The reference has been added when mentioning the study for the first time, in the same paragraph, so as not to confuse the reader (page 13, line 325).
Point 20. L283, please merge these paragraphs properly.
Response 20: The discussion has been reorganized for better reading and understanding.
Point 21. L298, “A possible explanation is that musicians with some complaints have been more motivated to answer the questionnaires. Another explanation is the influence of recall bias.” This is a big problem and may make these results invalid. More reasonable explanations are needed.
Response 21: The discussion text has been modified talking about recall bias and a better explanation has been given on the interpretation of the data based on the study biases.
Point 22. L312-313, merge into the previous one.
Response 22: Thank you for the suggestion. The sentence has been included in the previous paragraph (page 14, line 376).
Point 23. References, please give the DOI.
Response 23: Thank you. DOI has been added to references.

Round 2
Reviewer 1 Report
i want to congratulate the authors with an interesting study. The authors have taken notice of the comments and changed the manuscript accordingly. I have no further comments accept that the authors might look over the English in the revised version.
Author Response
Thank you for your comments. English has been revised by an English native speaker.
Reviewer 2 Report
The manuscript has been well revised and corrected based on my concerns and is recommended to be published in ijerph.
Author Response

(The authors gave the same response as above.)
